# NeuralTouch: Leveraging Implicit Neural Descriptor for Precise Sim-to-Real Tactile Robot Control

Anonymous Authors (Paper under double-blind review)

## I. INTRODUCTION

A commonplace behaviour in humans is our ability to glance at an object to determine its general position and then use touch alone to grasp it with precision. For example, after seeing where a plug is located in a socket, we can perform the precise manipulations to unplug and replug it using just our sense of touch. This behaviour remains challenging to implement artificially in robotics, and typically involves two phases: 1) an initial coarse phase where vision captures global information essential for contact-rich downstream tasks, and 2) and a subsequent fine phase where touch determines the optimal grasping pose, utilizing the prior visual information about the object's pose and geometry.

However, current applications of vision and touch for robotic manipulation tend to be constrained by several factors. First, in common scenarios objects are already optimally positioned in the hand for grasping by the robot [1]. Second, policies can be restricted to manipulating objects or contact features that were trained already, and so lack the ability to generalize to novel objects [2].Third, the independent use of vision and touch modalities can reduce their synergistic potential [3]. Fourth, multimodal policies developed in simulation struggle to transition seamlessly to real-world environments. This paper seeks to address these challenges by proposing a novel multimodal policy learning framework capable of overcoming these limitations.

In this work, we present *NeuralTouch*, a tactile RL policy learning framework with neural descriptor fields (NDF) [4]. Our goal is to improve the grasping accuracy of NDF-based methods with touch while maintaining sufficient generalizability to different inter-category objects. Furthermore, this framework does not restrict the NDF-based tactile servoing to limited, predefined contact geometries.

Experimentally, we focus on precise grasping with a tactile gripper through the aforementioned visual (coarse) and tactile (fine) phases. Specifically, in the coarse phase, we use an NDF to generate a pre-grasping pose, then the fine phase focuses on in-hand tactile servoing of the gripper fingers that repositions and reorients the gripper to achieve a specific grasp. This process is challenging due to the need to interpret the underlying object geometry in combination with precise control of a 6-DoF robot arm and parallel jaw gripper.

The main contributions of this work are as follows:
1) We propose a deep-RL-based framework with neural de-

All authors are with the Department of Engineering Mathematics and Bristol Robotics Laboratory, University of Bristol, Bristol BS8 1UB, U.K. (email: {yijiong.lin, n.lepora}@bristol.ac.uk)

scriptor fields to train a general tactile policy which does not need any explicit assumption about prior contact geometry.
2) We demonstrate that our NeuralTouch strongly complements state-of-the-art vision-based grasping to achieve the desired grasping pose with improved accuracy.
3) We validate this experimentally with zero-shot sim-to-real policy transfer and few-shot demonstration to showcase that our method solves a variety of downstream manipulation tasks over a variety of objects.

## II. METHODS

We separate the robotic grasping task into two phases: a coarse vision-guided phase and a fine tactile-guided phase. Note that while we structure this task similarly to other coarse-to-fine approaches [5], we do not rely on any specific methods from those approaches. In the coarse phase, we leverage the descriptor generated from NDF to calculate the coarse target grasping pose. Then, in the fine phase, we apply a tactile RL policy to accurately grasp an object with a desired contact pose represented by the NDF descriptor.

Specifically, we focus on learning a tactile RL policy that can be generalized to different target contact poses for different objects or tasks with the help of implicit neural descriptors from NDF. The tactile RL policy should not only consider the local contact to achieve safe, gentle contact but also have a sense of its desired contact pose with respect to the global shape of an object. Our method consists of three modules:
1) A PointNet Encoder [6] with Neural Pose Descriptor Fields [4] that learns implicit descriptors for various object shapes. These implicit representations describe the geometric relationships between poses (local frames) and the corresponding local shapes of inter-category objects.
2) A module to generate an initial coarse grasping pose using regression over the NDF [4].
3) A NeuralTouch RL module that learns a general tactile robotic policy conditioned on the implicit neural descriptors to achieve the desired fine grasping pose while maintaining safe, gentle physical interaction between the tactile robot and a manipulated object, given tactile and proprioceptive feedback.

## III. EXPERIMENTS AND RESULTS

Our experiments are designed to demonstrate the effectiveness and generalizability of the proposed NeuralTouch framework. In particular: 1) Are the NDF descriptors informative enough for one single RL policy to learn to grasp different contact features of different objects? 2) Does NeuralTouch improve the grasping accuracy on various unseen objects

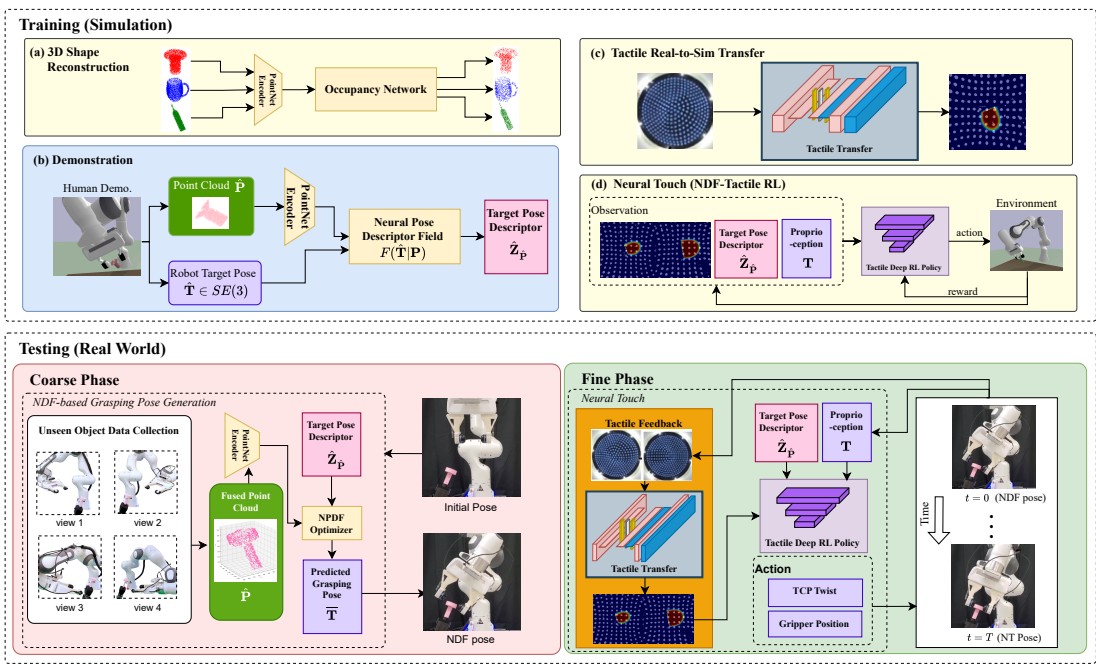

Figure 1. Overview of the NeuralTouch: In simulation, we first pre-train an occupancy network which is the core component of the Neural Pose Descriptor Fields. Secondly, we collect human demonstrations along with object point clouds and robot target grasping pose descriptors depending on the manipulation tasks. Thirdly, we train an RL policy with tactile and proprioceptive feedback, to achieve fine grasping poses implicitly specified by these collected descriptors. After obtaining the NPDF and a well-trained policy, our system is directly deployed in the real world with a real-to-sim tactile transfer to accurately grasp unseen objects, executing manipulation tasks such as unplugging a bolt-like USB and inserting it into a socket.

compared to the NDF-based grasping method on contact-rich manipulation tasks requiring high accuracy? 3) What is the NeuralTouch zero-shot sim-to-real performance?

### A. Tasks Setup

First, we design an ablation study and compare our method to two baselines: NDF [4] and NDF+RL-Touch. Specifically, we analyze grasping accuracy by measuring position errors and orientation errors for various target features of different objects in simulation. To further evaluate our proposed method, we consider two tasks both in simulation and in reality, with three phases: a) a coarse phase where the robot uses vision to locate and approach the unknown target feature pose of an object, b) a fine phase where the robot uses in-hand tactile servoing of the gripper fingers to achieve a precise grasping pose, and c) a replay phase where the robot executes a predefined skill to complete the task. The fine phase is particularly challenging due to the need for 7-DoF robot control and an understanding of the object's geometry to reposition and reorient the robot.

*1) Simulation Tasks:*

- Object Pick-and-Place: The robot aims to grasp a mug by its rim or horizontal handle, or a bottle by its neck, and place it horizontally on a table. The task is considered successful if the robot successfully picks up the object and places it on the table in a stable upright position. The object is initialized with a random pose above the table.
- Bolt out/in Hole:The robot aims to locate and unplug a bolt from one hole and insert it into another. The poses of

the bolt and the first hole are both unknown and initialized randomly above the table, while the pose of the second hole mounted on the table is known.

*2) Real-world Tasks:*

- Real Bottle Lid Opening: The robot aims to locate a bottle placed above the table and open its lid. The opening action of rotation is given by demonstration, hence the robot needs to grasp the lid with the correct pose to successfully execute the downstream task.
- This task is similar to that in simulation but with three out-of-distribution objects of increasing difficulty due to higher peg-hole clearance: a bolt, an adaptor plug, and a USB with a bolt-shaped adaptor. The yaw angle for the plug and USB is fixed, and the other components of their initial poses are sampled randomly.

### B. Ablation Study

We trained two policies (RL-Touch and NeuralTouch) in simulation for 6 target features of three kinds of objects: the mug rim, the mug wall, the mug right-angle handle, the mug horizontal handle, the bottle lid, and the bolt head. The RL-Touch policy achieves lower performance compared to the NeuralTouch policy, which we attribute to the lack of visual information leading to ambiguity when relying on touch only. Thus, the NeuralTouch policy achieves optimal performance on different target features compared to the baselines, which proves the effectiveness and efficiency of considering the implicit descriptor from NDF.

While the vanilla NDF approximately reach the target pose, they perform poorly due to the lack of fine-grained tactile

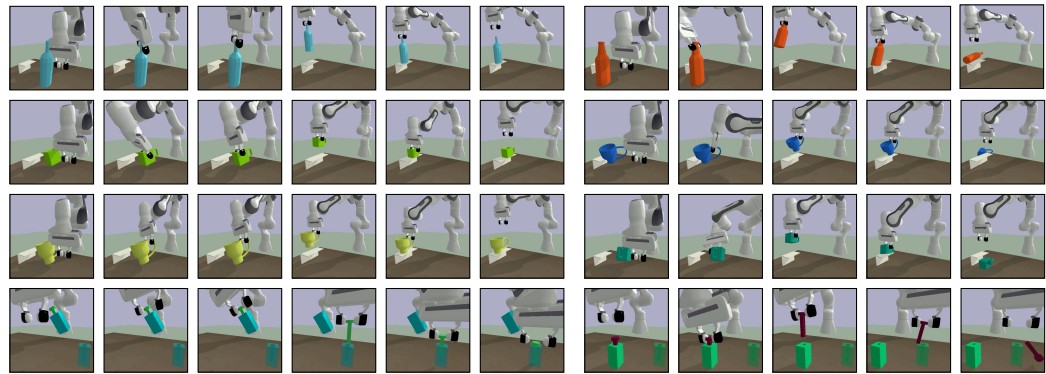

Figure 2. The snapshots of the robot performing four different tasks in simulation with three methods: (a) NeuralTouch, (b) NDF (first two rows) and NDF+RL-Touch (last two rows). From top to bottom row: object-pick-and-place (mug rim, mug horizontal handle, and bottle lid) and bolt-out/in-hole.

feedback. Interestingly, while the NDF+RL-Touch perform better than the vanilla NDF in reaching the bottle lid and the bolt head features, they perform similarly or worse for the other four target features. This is because the policy cannot distinguish which features are the targets, without the visual information from NDF, as the mug rim and mug wall (also the right-angle handle and horizontal handle) share many common tactile features during interactions, leading to ambiguity. In contrast, our NeuralTouch considers both modalities during the physical interactions, achieving significantly higher grasping accuracy.

### C. Simulation Tasks Results

We further evaluated the performance of the above methods in several fine manipulation tasks within simulation. Specifically, these are object pick-and-place and bolt-out/in-hole tasks, with 60 trials for each target feature. Our NeuralTouch method consistently outperforms the others in both tasks, because of its high accuracy.

For the bolt-out/in-hole task, while the vanilla RL-Touch policy seems to grasp the bolt with a desired pose, it more commonly reaches an inaccurate grasping pose, which can lead to insertion failures and thus a lower success rate compared to our NeuralTouch method. In the object pick-and-place task, although the vanilla RL-Touch policy performed slightly better than the vanilla NDF, its success rate remains far lower than NeuralTouch. This is because the RL-Touch policy suffered from an ambiguity in the target features. Conversely, by using NDF, our NeuralTouch method can avoid such ambiguities and achieve satisfactory performance.

### D. Real-world Tasks Results

In the bottle-lid-opening task, our NeuralTouch method achieves 90% success rate for the bottles of apple juice and the ketchup, and achieves 85% for the syrup bottle. In comparison, the vanilla NDF only achieved success rates of 30–45%. Thus, without tactile feedback, the NDF method frequently fails to open the bottle lid, as the rotation action must be executed precisely around the central axis of the cylinder-shaped lid to be successful. Also, when the gripper approached the lid with a large positional offset (where one finger was much closer

to the lid), the lid would oscillate forwards then back rather than continuously turn. These behaviours are shown in the supplementary video.

Both our method and the baseline were less successful when opening the syrup bottle lid, which we attribute to the smaller/shorter size of the lid. A failure case of NeuralTouch was that it sometimes could not locate the desired grasping pose during tactile servoing on very light contacts due to the real-to-sim tactile transfer (see supplementary video). Specifically, the real-to-sim tactile transfer can sometimes generate incorrect simulated images for light contacts because the marker motions are too subtle for the tactile transfer.

In the peg-out/in-hole task, our NeuralTouch method achieved success rates of 55%, 25% and 15% for the bolt, plug and USB objects, respectively, consistent with the clearances of these objects progressively decreasing. Note that even though the success rates of NeuralTouch with the plug and the USB are not high, it does succeed sometimes, and this is a task in which the actions can be repeated until it succeeds. Therefore, another way to interpret the results is that they take a longer times to complete. Also, even when the task fails on the insertion, it only has about 1 mm error, compared to clearances of 1 mm and 0.5 mm respectively.

In contrast, the baseline (vanilla NDF) barely succeeded with the bolt (5%) and consistently failed with inserting the plug and USB. The primary cause of failure was the misalignment of the grasping pose for insertion. Additionally, we observed that most failures with the bolt and the USB were due to misalignment between their main axes and the grasping pose, which resulted in significantly higher frictional resistance and prevented them from being withdrawn from the hole.

### IV. CONCLUSION

We presented NeuralTouch, a new method to achieve accurate robotic grasping that integrates vision and touch to enable precise manipulation with various objects and target features of those objects. Our approach consists of two main phases: a coarse phase, where the NDF is used to generate an initial grasping pose; and a fine phase, where the robot engages in tactile servoing using a neural descriptor-based RL tactile policy upon approaching the initial pose. Additionally,

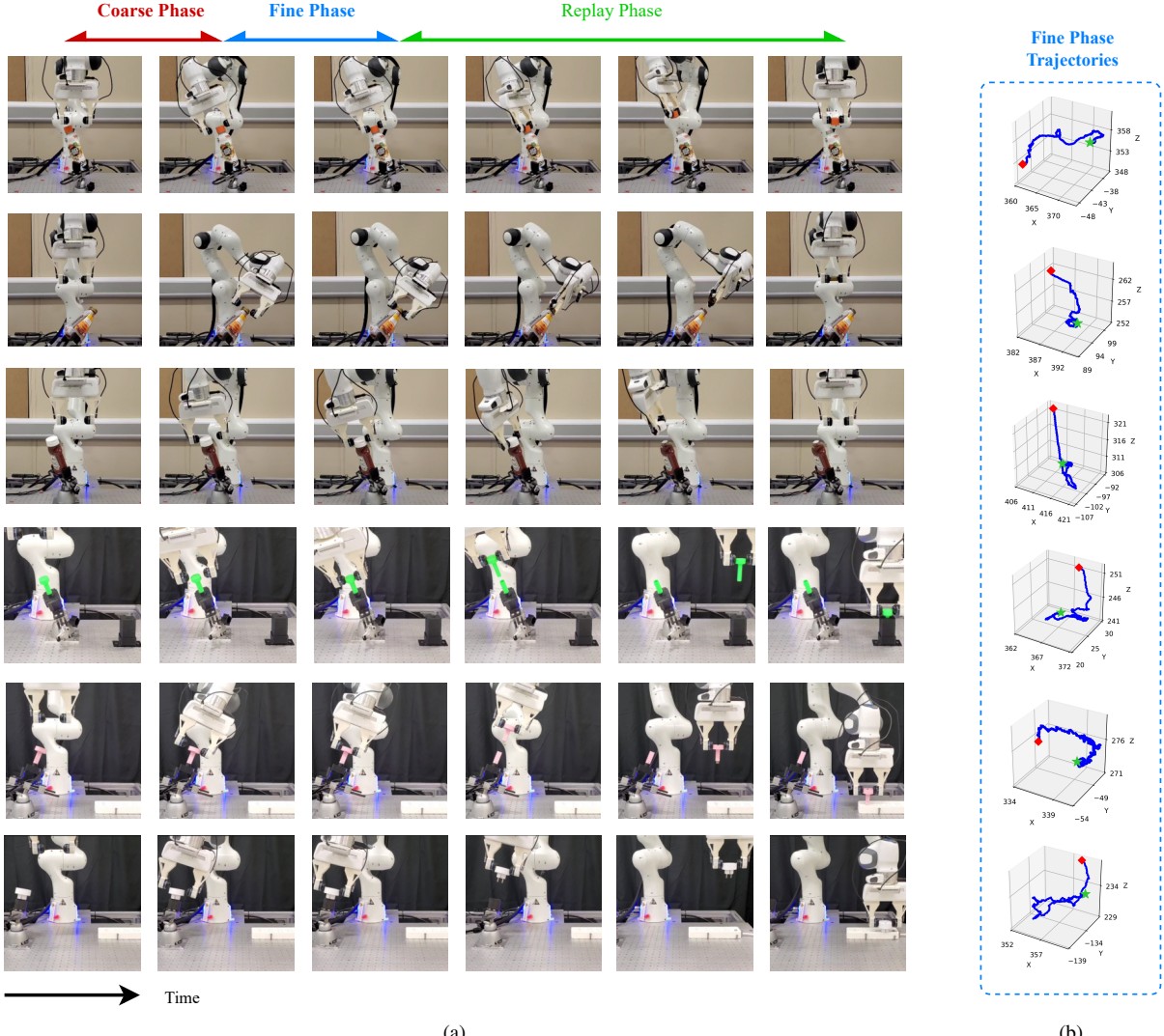

(a)          (b)

Figure 3. (a) Robot arm equipped with a tactile gripper performing two real-world manipulation tasks requiring high accuracy. Top 3 rows: bottle-lid opening. 4th row: peg-in/out-hole insertion. Bottom 2 rows: to increase the difficulty of the insertion task, we also experimented with a USB-head bolt and a plug where the clearances were approximately 0.5 mm and 1 mm, respectively. (b) End-effector trajectories recorded during the second phase. The red diamond represents the initial position determined by the NDF, while the green star indicates the final position achieved after tactile servoing.

we demonstrate applications of our method by introducing a third replay phase, where the robot performs downstream tasks requiring high precision, such as peg-out/in-hole. Our ablation study shows that NeuralTouch significantly outperforms baseline methods in grasping accuracy and generalizability. Furthermore, our method is sim-to-real transferable, which makes it easy to deploy in real-world scenarios.

While references [2] and [7] are similar to ours in this regard, they differ in explicitly predicting only the in-hand contact pose for known object shapes; thus, they do not perform real-time closed-loop control and are unable to react to external disturbances or generalize to unseen objects. Our method implicitly learns to achieve the desired grasping pose for various unseen objects.

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
