# OpenReview forum: "NeuralTouch: Leveraging Implicit Neural Descriptor for Precise Sim-to-Real Tactile Robot Control"
_IEEE.org/ICRA/2026/Workshop/Manipulation_Robustness — ICRA 2026_

### Official Review · Reviewer_NgNs · 2026-05-14
**Coarse-to-Fine Pose Refinement for Precise Manipulation**

**Rating:** 6
**Confidence:** 3

**Review:**

This paper presents NeuralTouch, a visuotactile coarse-to-fine manipulation framework that combines NDF-based coarse grasping with tactile RL-based pose refinement. The system is evaluated in simulation and real world tasks.

Strengths

- The motivation for integrating vision and touch for precise manipulation is timely and well grounded. The coarse-to-fine decomposition aligns intuitively with how humans combine these modalities.

Weaknesses

- The technical novelty appears moderate, with the main contribution residing in system integration and empirical validation rather than new methodological insights.
- The connection between the fine pose and the replay phase is not sufficiently explained. The paper would benefit from discussing how the refined pose enables successful replay, and how sensitive the replay outcome is to small grasp-pose deviations.
- Relying on a predefined replay phase somewhat tempers the generalizability claim. As written, the framework appears to generalize primarily at the grasp-refinement stage rather than across full manipulation behaviors.
- The real-world setup is somewhat simplified by fixing the yaw angle for the plug and USB tasks. Since this reduces the pose-estimation difficulty, the reported success rates may not fully reflect performance under more realistic conditions. A clearer explanation of this setup would also help avoid potential confusion.

---

### Decision · Program_Chairs · 2026-05-21

Accept